# New Efficient Method for Hysteroscopic Isthmoplasty: Four Simple Steps Lead to a Significant Improvement in Bleeding Status

**DOI:** 10.3390/jcm11216541

**Published:** 2022-11-04

**Authors:** Chien-Chu Huang, Shao-Chih Chiu, Chih-Ming Pan, Chun-Chung Huang, Cherry Yin-Yi Chang, Shih-Chi Chao, Der-Yang Cho, Wu-Chou Lin

**Affiliations:** 1Graduate Institute of Biomedical Sciences, China Medical University, Taichung 40402, Taiwan; 2Department of Obstetrics and Gynecology, China Medical University Hospital, Taichung 404327, Taiwan; 3Translational Cell Therapy Center, China Medical University Hospital, Taichung 404327, Taiwan; 4Institute of Oral Sciences, Chung Shan Medical University, Taichung 40201, Taiwan; 5School of Medicine, China Medical University, Taichung 40402, Taiwan; 6Department of Medical Research and Education, Lo-Hsu Medical Foundation, Lotung Poh-Ai Hospital, Yilan 265501, Taiwan; 7School of Chinese Medicine, China Medical University, Taichung 40402, Taiwan

**Keywords:** isthmocele, hysteroscopy, uterine scar defect, hysteroscopic isthmocele repair, postmenstrual spotting

## Abstract

We demonstrate an effective reduction in postmenstrual spotting after our novel hysteroscopic isthmoplasty. This study included 66 patients with isthmocele-related postmenstrual spotting confirmed by sonography and diagnostic hysteroscopy between 2000 and 2017. Our new interventions included the following four steps: (1) make a resection gradient of the distal edge of the isthmocele from the ape of the isthmocele down to the cervical outer orifice; (2) resect the distal and proximal niches of the isthmocele; (3) electrocauterize the distal and proximal sides (not only the niche bottom) of the small cave on the scar side of the isthmocele; (4) manage the isthmocele until it is largely connected to the cavity. In our results, all patients underwent extensive hysteroscopic repair of newly hysteroscopic isthmoplasty without any intra- or postoperative complications. After final hysteroscopic repair modification, prolonged menstrual spotting was significantly decreased in 98.2% (53/54) of the patients, and the total number of bleeding days per menstrual cycle significantly decreased from a mean of 15.38 ± 3.3 days to 6.4 ± 1.9 days postoperatively (*p* < 0.001). Our four-step hysteroscopic technique successfully resolved prolonged menstrual spotting in over 90% of the patients, exceeding the resolution rates of 60–85% achieved with other hysteroscopic techniques used to treat symptomatic isthmocele. No patients experience recurrence after long-term follow up. Four simple steps led to a significant improvement in bleeding status.

## 1. Introduction

A cesarean section is the most commonly performed surgical procedure among women worldwide [1]. Cesarean incisions made in the isthmus or the lower uterine segment can result in weakness and fibrosis of the uterine wall, identifiable by sonography or hysteroscopy [2,3]. A uterine isthmocele is a niche or defect that forms in the anterior wall of the lower uterine segment at the site of a previous cesarean scar [4].

Isthmoceles occur in 56% to 84% of cesarean deliveries [5,6]. The formation of a defect is associated with premature rupture of membranes, short operation time, cervical dilatation during or before a cesarean section, or multiple cesarean sections [7]. While isthocele may be asymptomatic, symptoms of isthmocele include abnormal uterine bleeding, dyspareunia, pelvic pain, dysmenorrhea, and infertility [4]. Several imaging methods are used to diagnose isthmoceles [8].

In the out-patient department, the most common symptom of isthmocele is prolonged bleeding for more than 2 weeks, without an increase in the total bleeding amount of menstrual bleeding. Although medical control is on offer, most of our patients who underwent surgery had already tried long-term remedies including oral contraception pills, uterine contraction medication, or tranexamic acid, to no effect. The symptoms of isthmoceles adversely affect quality of life and require surgical intervention for correction. Isthmoceles are repaired by laparotomy, hysteroscopy, laparoscopy, or vaginal surgery [9,10]. Of these, hysteroscopic surgery is the most cost effective [11], but typically only about 60–66% of patients report an improvement in symptoms postoperatively [12,13]. New, efficient, and minimally invasive surgery is required. This study demonstrates a novel four-step isthmoplasty that significantly improves patients’ bleeding status over a 17-year period.

## 2. Materials and Methods

### 2.1. Study Design

This retrospective study included 66 women with prolonged menstrual spotting due to isthmoceles who had undergone hysteroscopic repair between January 2000 and December 2017 in China Medical University Hospital (CMUH, Taichung, Taiwan). The study was approved by the Institutional Review Board of CMUH (IRB No. CMUH 108-REC1-095). All surgeries were performed by Dr. William Wu-Chou Lin. All patients were counseled on the excision of coexisting findings such as ovarian cysts, endometriosis, and adhesions.

A preoperative diagnosis of an isthmocele was established using the patients’ histories of symptoms and was visualized for all patients using sonography. A triangular defect over the anterior uterine wall accompanied by fluid accumulation was noted under sonography. All diagnoses were confirmed by diagnostic hysteroscopy. If the distance of the isthmocele to the bladder was less than 4 mm under sonography, another surgical approach was employed.

### 2.2. Performed Surgery

Our operation device was resectoscope of Richard Wolf (26 Fr., 4 mm, 0°) with bipolar electrode Loop. The novel procedure concentrates on four aspects: (1) the granulation trap inside the isthmocele; (2) fibrotic tissue over the proximal side of the isthmocele; (3) the small caves in the isthmocele scar; and (4) the isthmocele itself. These are the four main sources of blood coagulation (Figure 1 and Figure 2a) and must be addressed to reduce prolonged bleeding. The procedure involves performing an initial resection gradient of the distal edge of the isthmocele from the ape of the isthmocele down to the cervical outer orifice. Secondly, the distal and proximal niches of the isthmocele are resected. Thirdly, the small cave on the scar side of the isthmocele is electrocauterized on the distal and proximal sides, not only over the niche bottom. Fourthly, the isthmocele is managed until it is largely connected to the cavity (Figure 2). Isthmocele defects can also occur at more than one site. In our study population, three defects were found in one patient who had undergone three cesarean sections.

### 2.3. Follow-Up

The initial Outpatient Department (OPD) follow-up was at 1 week postoperatively for management of acute postoperative complications. The second OPD follow-up was at 1 month postoperatively. If there were no significant complaints, the follow-up period was changed to 1 year. Each patient was followed-up for at least 4 years. At each follow-up, patients underwent sonography imaging and were asked about the resolution of symptoms based on their personal impression (“improvement of symptoms” vs. “no improvement”). Complete resolution was defined as menses lasting ≤10 days.

### 2.4. Statistical Analysis

We used SAS software version 9.4 (SAS Institute, Cary, NC, USA) for statistical analysis. Categorical variables are shown as numbers and continuous variables are shown as means ± 1 standard deviation. The paired *t*-test was used to compare preoperative and postoperative bleeding day characteristics for all patients. Student’s *t*-test was applied in the comparison of outcomes for the study groups before and after modification of the surgical procedure. A *p*-value of <0.05 was considered statistically significant.

## 3. Results

### 3.1. Clinical Characteristics of the Patients

The clinical characteristics of all 66 patients are presented in Table 1. The mean age at the time of surgery was 38.7 years. Sixty patients (60/66, 90%) had a history of more than two cesarean sections. All patients presented at the OPD with prolonged menstrual spotting. Other symptoms included dysmenorrhea (12/66, 18%), pelvic pain (6/66, 9%) and lower back pain (2/66, 3%). The mean duration of prolonged menstrual spotting before surgery was 5.84 ± 4.2 years.

### 3.2. Magnificent Modification of Procedure

According to the data from our learning curve, we can separate the patients into two groups: the first 12 cases using the unmodified four-step method, and the other 54 cases under the final modified four-step method (Figure 3).

Before modification of the procedure, the total bleeding days per menstrual cycle improved from 13.2 ± 2.9 days preoperatively to 7.0 ± 2.0 days postoperatively; after modification, the corresponding improvement was from 15.38 ± 3.3 days to 6.4 ± 1.9 days (*p* < 0.001) (Figure 3a).

The operating procedure of the first 12 cases involved just a small resection of the distal edge of the isthmocele (Figure 2b, white line) that did not extend to the cervical outer orifice. Under clinical response status, the first step of the procedure was therefore modified to resect the distal edge of the isthmocele down to the cervical outer orifice in a long gradient (Figure 2b, blue line).

### 3.3. Significant Improvement of the Clinical Successful Rate

All patients underwent hysteroscopic isthmoplasty hysteroscopy without any intra- or postoperative complications. After hysteroscopic isthmoplasty, 61 patients (92.4%) experienced a significant decrease in prolonged menstrual spotting.

According to the data from our learning curve, the period of prolonged bleeding improved in the first 12 cases (from 13.25 ± 2.9 days to 7 ± 2 days), but the bleeding duration was still not short enough (Figure 3). Nine (75%) of the 12 patients achieved complete resolution; in the remaining three patients, prolonged bleeding was reduced by only 4 days (from 14 to 10 days).

Using the modified four-step procedure incorporating a long slope cutting tract, only 1 out of 54 patients experienced prolonged menstrual spotting for more than 10 days postoperatively, and the issue was resolved completely in 98.2% of patients; a significantly superior outcome compared with the results obtained using the initial hysteroscopic technique (*p* = 0.0266) (Figure 3b). In a comparison of outcomes from the two groups (N = 12 vs. 54), the modified procedure decreased the bleeding period by a mean of 8.9 ± 1.4 days compared with only 6.29 ± 0.8 days for the previous method (*p* = 0.0266) (Figure 3b).

### 3.4. Minimal Invasion with Less Blood Loss and Short Admission

The mean operation time was 66 ± 39 min (range 30–215 min) and mean blood loss was 10 mL (range 5–50 mL). The average stay in hospital was 3 days. The median follow-up of the patients was 8.5 years (4–15 years). Patients were checked once a year with sonography during an OPD visit. Our records revealed no recurrence of symptoms for any of the patients.

## 4. Discussion

The incidence of isthmocele has steadily increased as the number of caesarean sections performed worldwide has increased over the past few decades [3]. Isthmoceles are treated using laparotomy, hysteroscopy, laparoscopy, or vaginal surgery [14]. Studies have shown that hysteroscopy effectively treats prolonged spotting due to isthmoceles [15,16]. A recent systematic review and meta-analysis of the efficacy and safety of different surgical approaches for the treatment of symptomatic isthmocele found that hysteroscopic correction improved symptoms in about 85% of the patients and was associated with the lowest risk of complications [8,17]. The literature has discussed important concepts regarding the nature of surgical techniques for the treatment of symptomatic isthmocele: whether to resect one or two sides of the defect, and how to manage the defect itself [1,17,18,19]. As yet, no consensus has been reached as to the most appropriate methods for hysteroscopic treatment of isthmocele [1,8,12,18,19,20,21,22,23,24,25,26] (Table 2). Clinical improvement has been noted between 59.6% and 100% [12,26], but there has been no consistent definition for improvement of outcomes during postoperative follow-up [4]. Moreover, comparing previously reported clinical outcomes of surgical procedures for treating symptomatic isthmoceles is hindered by the small patient samples, with numbers of enrolled patients ranging between 18 and 57; the largest-ever patient cohort is 120 (Table 2) [1,8,12,18,19,20,21,22,23,24,25,26].

In this study, using the final modified four-step procedure incorporating a long slope cutting tract, prolonged menstrual spotting was resolved completely in 98.2% of patients (Figure 3a,b). The key point of this procedure is that the operation gradient extends from the ape of the isthmocele to the outer cervical orifice, preventing the creation of a new, larger cave capable of trapping blood and granulation tissue. Both proximal and distal niches of the isthmocele must be resected to achieve total resolution, or the residue side would still trap the blood. Additionally, women can have two different isthmoceles in parallel, and the possible second defect is also corrected when the resection of the proximal niches is performed. Our technique enables the isthmocele to be largely connected to the cavity. A video of this final procedure has been published [14]. A limitation of hysteroscopic surgery is that this technique is incapable of strengthening the uterine wall [28]. If the patient wants to bear children in future, our hospital uses a laparoscopic approach, to avoid subsequent pregnancy-related complications [28]. Laparoscopic repair using sutures allows good anatomic recovery [29,30]. Robotic-assisted repair of the isthmocele is also an effective and safe method [31,32], but the duration of the surgery, the degree of blood loss, and the length of hospital stay are all greater than those with a hysteroscopic approach [31].

Another important consideration is the anatomy of the defect. Previous studies have claimed isthmocele in the lower segment of the uterine cavity [33,34]. However, in all our patients, the scar defect appeared to be located over the internal orifice side of the cervical canal and close to the squamocolumnar junction, not over the low segment of the uterine cavity. Though there are no references yet, it is critical to address this concept with relevant experts. Notably, as all 66 patients in our study exhibited the same anatomic conditions in the operating theater, our clinical findings strongly support the contention that the defect was located over the internal orifice side of the cervical canal.

Although hormonal modulation is effective for isthmocele, hysteroscopic excision is an effective strategy for patients who are hesitant or concerned about long-term medical needs [35]. We provide individualized care in our professional practice, which includes giving patients all treatment alternatives, such as oral contraceptives, a levonorgestrel intrauterine device, follow-up alone, or surgery.

Other comorbidities that may cause prolonged bleeding and menorrhagia should therefore be evaluated prior to surgery include uterine comorbidities, such as uterine myomas and adenomyosis. One patient who experienced limited improvement after surgery had existing adenomyosis and received hormonal treatment for menorrhagia for several years following hysteroscopic isthmoplasty. This patient finally received LTH due to menorrhagia and anemia. Moreover, a scar sited closer to the bladder side prevented access to a defect that was too deep. Interestingly, we found more than one scar defect in some of the patients during surgery. Even with more the one scar of defects were noted, following our four steps operation method, total resolution can achieve.

A potential limitation of this study is the nature of the retrospective study design, which can introduce important sources of bias including loss to follow-up and methods of follow-up that can result in real or perceived bias in the reporting of study outcome(s) [36,37]. Some specific clinical factors, like the number of defects, the use of medication before operation, and contraception after operation, may not be able to be included on the basis of previous medical records. However, our study clearly quantified an improvement in clinical outcome (expressed as fewer bleeding days). Furthermore, as all operations were performed by the same surgeon in the same hospital, this eliminated any possibility of cognitive bias in surgical performance between surgeons and hospitals. Additional strengths of the retrospective design study include the total patient number of 66 and the long follow-up period of over 48 months for each patient, with also a long study span of about 17 years, allowing us to gather experience and have sufficient time to modify and test an alternative hysteroscopic technique. However, the study did not include patients planning future pregnancies. Thus, potential long-term complications such as the management of uterine rupture or preterm labor in subsequent pregnancies did not have to be analyzed.

The hysteroscopic technique detailed in our study not only resulted in clinical resolution of symptoms in over 90% of the patients, the modified approach also exhibited a higher clinical efficacy compared with previous hysteroscopic techniques [8,12,13].

## 5. Conclusions

Our new four-step hysteroscopic approach to isthmocele repair enables patients to be discharged early with minimal blood loss and a high rate of clinical success up to 90%. Importantly, this modified procedure allows patients to achieve a good quality of life without long-term medication need.

## Figures and Tables

**Figure 1 jcm-11-06541-f001:**
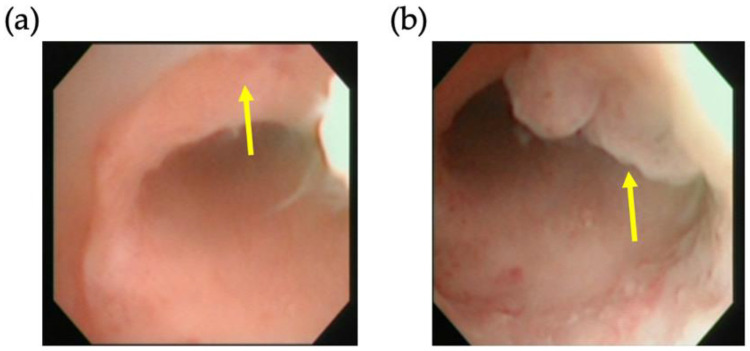
Hysteroscopic view of a representative isthmocele. Arrows mark (**a**) the defect and (**b**) the granulation trap inside the isthmocele.

**Figure 2 jcm-11-06541-f002:**
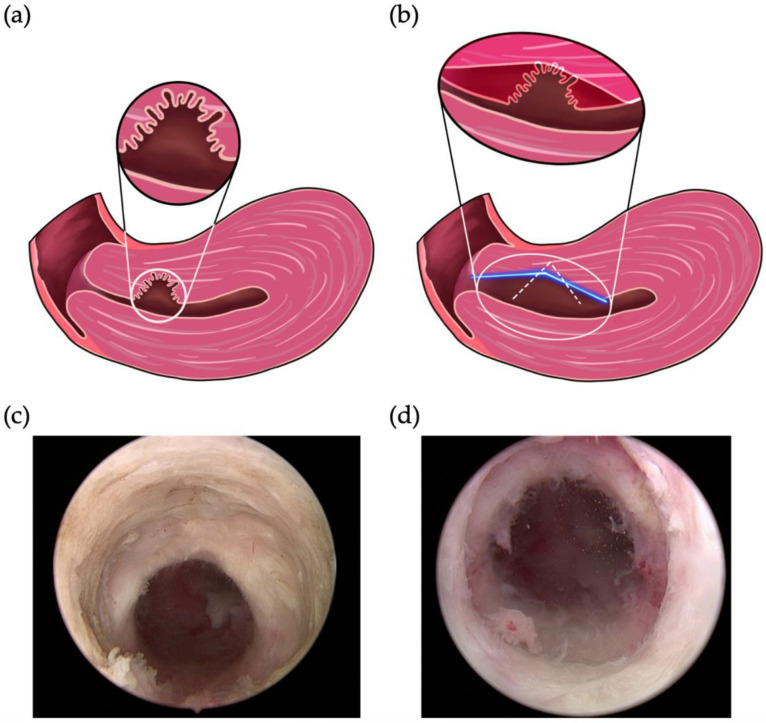
Our novel four-step approach to surgical hysteroscopic treatment of cesarean-induced isthmoceles. Schematics of (**a**) per-operation status of isthmoceles and (**b**) the resection of the distal edge of an isthmocele using two methods. The white dotted line and blue line indicate the position of the resection using the previous and final modified method, respectively. Representative postoperative image of the repaired (**c**) inner and (**d**) outer canal.

**Figure 3 jcm-11-06541-f003:**
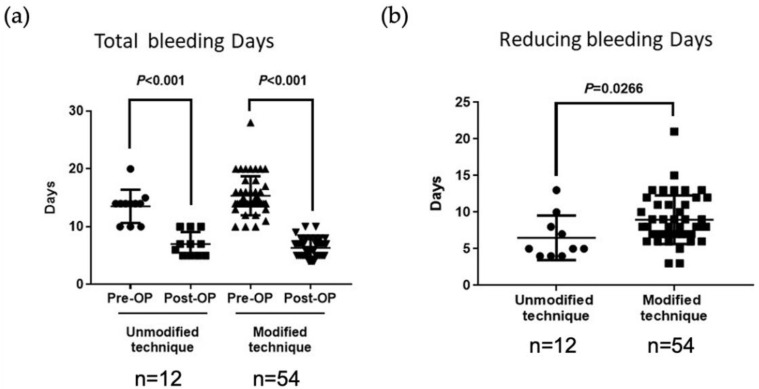
A comparison of preoperative and postoperative total number of bleeding days per menstrual cycle in patients with isthmoceles treated with a four-step hysteroscopic technique. (**a**) Before the technique was modified (n = 12), the total bleeding days per menstrual cycle improved from a mean 13.2 ± 2.9 days preoperatively to 7.0 ± 2.0 days postoperatively. After modifying the technique (n = 54), the total bleeding days per menstrual cycle improved from a mean 15.38 ± 3.3 days preoperatively to 6.4 ± 1.9 days postoperatively. *Pre*, preoperatively; *Post*, postoperatively. (**b**) A comparison between the unmodified (n = 12) and modified (n = 54) techniques revealed that the new method decreased the bleeding period by 8.9 ± 1.4 days versus only 6.29 ± 0.8 days with the previous method (*p* = 0.0266). *Pre*, preoperatively; *Post*, postoperatively.

**Table 1 jcm-11-06541-t001:** Baseline characteristics of the study population.

Variable	Value
Age, years	38.7 (27–49)
Prolonged uterine bleeding	66 (100%)
Dysmenorrhea	12 (18%)
Pelvic pain	6 (9%)
Low back pain	2 (3%)
Mean duration of symptoms (years)	5.84 (1–17)
Number of previous caesarean sections	
1	6 (10%)
2	40 (60%)
3	20 (30%)
Duration of operation (min)	66 (30–215)
Blood loss (mL)	10 (5–50)
Mean follow-up (years)	8.5 (4–15)

Values are presented as the mean (range) or number of patients (%).

**Table 2 jcm-11-06541-t002:** Appropriate methods for hysteroscopic treatment of isthmocele.

References	Study Design	Numbers of Patients	Rates of Success (%)	Follow-Up Duration (months)	Pre-Op Bleeding Days	Post-Op Bleeding Days	Mean Operating Time (min)	Operative Blood Loss (mL)	Concept of Operations
Our study									
Overall	Retrospective	66	92.4	102	15 ± 3.37	6.6 ± 2	66	10	I, II, III, and IV
Unmodified		12	75		13.2 ± 2.9	7.0 ± 2.0			(I, not to the cervical outer orifice)
Modified		54	98.2		15.38 ± 3.3	6.29 ± 0.8			(I, to the cervical outer orifice)
Other studies									
[20]	Retrospective	24	84	14 to 24	-	-	-	-	
[26]	Prospective	26	100	12 to 23	2 to 12	-	-	-	II and III
[12]	Retrospective	57	59.6	60	12.9 ± 2.9	9.4 ± 4.1	30.2 ± 6.6	-	II (distal side only) and III
[8]	Retrospective	57	66	12	9.6	6.6 to 7.6	-	-	II (distal side only) and III
[19]	Retrospective	24	79	3 to 16	-	-	23 ± 15	11 ± 6	II and III
[21]	Retrospective	22	68	-	-	-	-	-	II (proximal side only)
[24]	Retrospective	31	64.5	22.3	14	-	25	10	III
[22]	Prospective	120	80	35	-	-	8	-	II (distal side only)
[23]	Prospective	18	64	12	-	-	-	-	-
[27]	Prospective	19	N/A	21.2	11.9 ± 3.1	7.9 ± 2.2	25 ± 14	10.7 ± 16	II (distal side only)
[18]	Prospective	38	96.8	2	-	-	-	-	II (distal side only) III
[18]	Prospective	38	87.5	2	-	-	-	-	II and III
[1]	Prospective	23	68.8	6	7	-	17.4 ± 3.5	-	II and III

I—Make a resection gradient of the distal edge of the isthmocele from the ape of the isthmocele down to the cervical outer orifice; II—resect the distal and proximal niches of the isthmocele; III—electrocauterize the distal and proximal sides (not only the niche bottom) of the small cave on the scar side of the isthmocele; IV—manage the isthmocele until it is largely connected to the cavity.

## Data Availability

Data is contained within the article.

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
