# Peer review of "New Efficient Method for Hysteroscopic Isthmoplasty: Four Simple Steps Lead to a Significant Improvement in Bleeding Status"

_jcm, 2022, doi:10.3390/jcm11216541_

Round 1
Reviewer 1 Report
1) the title is A Highly Successful Four-Step Hysteroscopic Surgical Tech-2 nique for Isthmocele Repair: Experience from a Single Medical 3 Center in Taiwan. A more direct title like Bledding pattern after four-step hysteroscopic isthmocele repair would be more honest. Most articles report infertility outcomes after hysteroscopic repair. In this case you report only the bleeding pattern because no fertility wish.
2) line 70: the usual described hysteroscopic treatment would be resection of the part of the isthmocele near the cervix and coagulation of the isthmocele itself. Your four step procedure seems to add only the resection of the part of the istmocele closer to the uterine cavity. What is the theory behind it?
3) results are not clear. line 121 - several group analysis ? first 12 patients had a different procedure?
4) line 140 - modification of the procedure. When? How many patients in each group?
5) line 154 - stay 3 days in the hospital? Why? In my practice is an outpatient procedure.
6) line 163 - please elaborate on the oral contraceptive treatment given because in the literature most patients would have been treated in a very successful way with oral contraceptives or even levonorgestrel intra uterine device.
7) line 204-5: please explain because it may be cause by to low uterine incisions in cesarean ? Is this described in the literature? Please compare and do not say "it would be interesting to discuss...".
8) line 210 : only one patient with hormonal treatment? The others did contraception by using what? Again this is an important cofunding factor that should be described in results and discussion.
9) line 226 - no pregnancies in these 66 patients ? please explain.
Reviewer 2 Report
Dear author,
Congratulations on your work; this work was fascinating. But regarding the new 4-step or new modified technique you want to show, it was not enough background to rationalize the need to modify the technique.
1. In the Introduction section, please describe the reason for doing the modified technique of isthmocele repair, including the urgency of changing it briefly.
2. At the end of the Introduction, the author did not mention the article's aim for the modification of the repair technique, so please note it (it was only about the clinical parameters associated with symptomatic improvement.
3. In the Materials and Methods, please also mention what resection devices were used to resect.
4. The result said that in most patients, you found more than one scar defect. Could you give the data on the baseline characteristics of the study population? Because this was also important in advocating that your new modified technique could be applied to multiple defects.
Round 2
Reviewer 1 Report
Interesting article. Two group inclusion weakness the article because is not so clear the goal, etcetera.
I would add a possible explanation for your results of upper edge of the isthmocele resection improving the results : women can have two different c-section scar defects parallel and when the resection of the upper edge is done the possible second deffect is also corrected.
